# Latanoprostene Bunod 0.024% in the Treatment of Open-Angle Glaucoma and Ocular Hypertension: A Meta-Analysis

**DOI:** 10.3390/jcm11154325

**Published:** 2022-07-26

**Authors:** Tzu-Chen Lo, Yu-Yen Chen, Man-Chen Hung, Pesus Chou

**Affiliations:** 1Department of Medical Education, Taichung Veterans General Hospital, Taichung 407, Taiwan; tzuchenlo@gmail.com (T.-C.L.); manchenhung@gmail.com (M.-C.H.); 2Department of Ophthalmology, Taipei Veterans General Hospital, Taipei 112, Taiwan; 3School of Medicine, National Yang-Ming University, Taipei 112, Taiwan; 4Department of Ophthalmology, Taichung Veterans General Hospital, Taichung 407, Taiwan; 5Wilmer Eye Institute, Johns Hopkins University, Baltimore, MD 21287, USA; 6School of Medicine, Chung Shan Medical University, Taichung 402, Taiwan; 7Department of Post Baccalaureate Medicine, National Chung Hsing University, Taichung 402, Taiwan; 8School of Medicine, Mackay Medical College, New Taipei City 252, Taiwan; 9Institute of Public Health, National Yang-Ming University, Taipei 112, Taiwan; pschou@ym.edu.tw

**Keywords:** latanoprostene bunod, open-angle glaucoma, ocular hypertension, meta-analysis

## Abstract

Latanoprostene bunod (LBN) 0.024%, a newly approved glaucoma eye drop, is metabolized into latanoprost acid and a nitric oxide (NO)-donating moiety, thus increasing the outflow of aqueous humor through the uveoscleral and trabecular routes, respectively. This study aimed to evaluate the intraocular pressure (IOP)-lowering effect of LBN among patients with open-angle glaucoma (OAG) and ocular hypertension (OHT). The effectiveness of LBN was also compared with timolol maleate 0.5% and latanoprost 0.005%. We searched PubMed and Embase between 1 January 2010, and 31 March 2022 and adopted only peer-reviewed clinical studies in our meta-analysis. A total of nine studies (2389 patients with OAG or OHT) assessing the IOP-reduction effect of LBN were included. Standardized mean differences (SMDs) of IOP between post-treatment time points (2 weeks, 6 weeks, 3 months, 6 months, 9 months, and 12 months) and baseline were calculated. The pooled analysis according to each time point revealed a significant IOP drop after LBN treatment (all *p* values for SMD < 0.05). In addition, LBN revealed a significantly stronger efficacy in decreasing IOP than timolol maleate 0.5% and latanoprost 0.005% during the follow-up period of three months. No serious side effects of LBN 0.024% were reported. Our study concluded that LBN could achieve good performance for IOP reduction in patients with OAG and OHT. The safety was favorable with no severe side effects.

## 1. Introduction

Glaucoma is the leading cause of irreversible blindness [1]. Lowering the intraocular pressure (IOP) can slow optic nerve injury and visual field deterioration and is thus a proven effective treatment [1,2]. To this end, clinicians have relied on glaucoma eye drops as a means to either reduce aqueous humor production or increase aqueous humor outflow facilities through trabecular meshwork or uveoscleral routes [3]. Beta-blockers, cholinergic agents, carbonic anhydrase inhibitors, alpha-adrenergic receptor agonists, and prostaglandin analogs have remained the standard of care for over 20 years.

Li et al. revealed that prostaglandin analogs (e.g., latanoprost 0.005%) and beta-blockers (e.g., timolol maleate 0.5%) are among the most efficacious categories of glaucoma eye drops in reducing IOP [4]. Although these therapies certainly have a positive impact, they are insufficient at hindering disease progression in 30–80% of glaucoma patients [2,5]. Therefore, medications with novel mechanisms of action are needed. 

Latanoprostene bunod (LBN) 0.024% is a new IOP-lowering eye drop approved by the US Food and Drug Administration (FDA) in November 2017. LBN is a nitric oxide (NO)-donating prostaglandin F2α analog that, when applied to the ocular surface, is rapidly metabolized into latanoprost acid and a NO-donating moiety, butanediol mononitrate [6,7]. Latanoprost acid reduces IOP by increasing uveoscleral outflow, whereas NO facilitates trabecular outflow through relaxation of the trabecular meshwork and Schlemm’s canal [8].

Randomized controlled trials (RCTs), such as APOLLO, JUPITER, LUNAR, VOYAGER, and CONSTELLATION, as well as a pooled study conducted by Weinreb et al., have assessed the IOP-reduction effect of LBN in patients with open-angle glaucoma (OAG) and ocular hypertension (OHT) [9,10,11,12,13,14]. Since the approval of LBN, retrospective studies (e.g., chart reviews) regarding its IOP-lowering effect in the real-world setting have been performed [15,16,17]. To ensure a complete understanding of LBN 0.024%, we investigated its efficacy in decreasing IOP in both the RCTs and retrospective studies. In this meta-analysis, we aimed to evaluate the impact of LBN 0.024% on IOP, which was further compared with timolol maleate 0.5% and latanoprost 0.005%. The adverse effects of LBN 0.024% were also recorded for a safety assessment.

## 2. Materials and Methods

### 2.1. Search Strategy

This study was conducted in accordance with the Preferred Items for Systematic Reviews and Meta-Analyses (PRISMA) guidelines. We performed a literature search in the PubMed and Embase databases for studies published from 1 January 2010 to 31 March 2022 using the keyword ‘latanoprostene bunod’. Papers were initially screened by examining the titles and abstracts. Then, the full texts were further assessed to determine whether the studies met the inclusion criteria. Bibliographies were also manually searched for the relevant literature.

### 2.2. Inclusion and Exclusion Criteria

We included only peer-reviewed journal articles written in English. Only original prospective studies or retrospective clinical studies were included. Studies met the following inclusion criteria: (1) study population of OAG or OHT patients; (2) a clear definition of the study design as well as the doses and frequency of eye drops; and (3) analysis of the effect of IOP lowering in LBN with or without comparison to timolol maleate 0.5% or latanoprost 0.005%. It was also noteworthy that patients included were either naive to glaucoma eye drops or had undergone a washout period to ensure that the previous prescription did not influence the effect of the study eye drop. Reviews, meta-analyses, and conference abstracts were excluded due to the inclusion of repeated cases. Two researchers (Lo and Chen) independently assessed the eligibility of these articles. Discordances were resolved by a third reviewer (Hung).

### 2.3. Extraction of Variables

The following data were recorded from the included articles: the first author, the year of publication, country, study design, glaucoma eye drops, total number of patients, total number of study eyes, age (mean and standard deviation) and sex distribution of participants, baseline IOP, and post-treatment IOP at each time point (1 week, 2 weeks, 6 weeks, 3 months, 6 months, 9 months, and 12 months). The adverse effects of LBN 0.024% were also listed and summarized.

### 2.4. Statistical Analysis

Comprehensive Meta-Analysis software, version 3 (Biostat, Englewood, NJ, USA) was used for the statistical analysis. The mean IOP reduction in the LBN at each post-treatment time point was calculated. For studies in which the LBN group was compared to timolol maleate 0.5% or latanoprost 0.005%, the differences in the IOP-lowering effect between groups were computed. Then, the standardized mean difference (SMD) of each study was derived by dividing the mean difference by the standard deviation to ensure that the difference was on the same scale. Then, the SMDs of the included studies were pooled to derive the overall differences at each time point. Forest plots were used to illustrate the point estimates of SMDs with a 95% confidence interval (CI). Moreover, the *I*^2^ statistic was used to assess heterogeneity across each trial. The *I*^2^ statistic reveals the percentage of variation between studies that is due to heterogeneity rather than chance or sampling error. An outcome of over 75% indicates considerable heterogeneity. Publication bias was determined using funnel plots and Egger’s test.

## 3. Results

### 3.1. Search Results

Figure 1 shows the PRISMA flow diagram of study screening. An initial search yielded 185 citations. After eliminating duplicated records (*n* = 46), 139 studies remained. Then, we excluded non-relevant studies (*n* = 68). Studies categorized as reviews, meta-analyses, and conference abstracts were also excluded (*n* = 56). We further excluded three animal studies, two papers not written in English, and one study in which LBN 0.024% was not used in glaucoma patients. Finally, nine studies were included in our meta-analysis.

### 3.2. Characteristics of Included Studies

Table 1 summarizes the characteristics of the nine studies. Seven studies were from the USA or Europe, and two were from Asia. Most of the studies were RCTs, and three studies were retrospective chart reviews. The study conducted by Weinreb et al. published in 2018 [14] was a pooled analysis of the LUNAR and APOLLO studies; thus, we only adopted data from the safety extension phase to avoid repetition. In total, 2389 patients were included in this meta-analysis with an average age of 63.2 years old.

### 3.3. Outcome Assessment

Table 2 shows the baseline and post-treatment IOP at each time point (1 week, 2 weeks, 6 weeks, 3 months, 6 months, 9 months, and 12 months) in the included studies. Seven studies had IOP data up to 6 weeks after treatment, and only one study had a follow-up time reaching 12 months. Four studies had one arm of LBN 0.024%, three studies (LUNAR, APOLLO, and CONSTELLATION) compared LBN 0.024% vs. timolol maleate 0.5%, one study (VOYAGER) compared LBN 0.024% vs. latanoprost 0.005%, and one study compared the effectiveness of all three drugs.

Figure 2 illustrates the SMDs of IOP between the post-treatment time points and baseline. All studies reported a significant reduction in IOP at every time point after LBN treatment. Subgroup analysis according to each time point revealed a significant pooled IOP reduction with LBN from 2 weeks to 12 months (SMD was −3.926 at 2 weeks, −2.429 at 6 weeks, −3.110 at 3 months, −2.081 at 6 months, −1.759 at 9 months, and −2.457 at 12 months).

Figure 3 compares the IOP-reducing effect between LBN 0.024% and timolol maleate 0.5% at 2 weeks, 6 weeks, and 3 months. At these three time points, LBN significantly reduced IOP more than timolol. The SMDs were −0.61 (95% confidence interval (CI): −0.95 to −0.27), −0.66 (95% CI: −0.81 to −0.52), and −0.98 (95% CI: −1.36 to −0.61), respectively.

Figure 4 compares the IOP reduction between LBN 0.024% and latanoprost 0.005% at 1 week, 2 weeks, 6 weeks, and 3 months post-treatment. After pooling the results of different time points, the overall SMD was statistically significant with a value of −0.599 (95% CI: −1.022 to −0.177).

Subsequently, we recorded the adverse events in each study to assess the safety of LBN 0.024%. The three most common ocular side effects were conjunctival hyperemia, eye irritation, and dry eye, as shown in Table 3. The percentage of patients who had at least one adverse effect ranged from 8.7% to 50.8%. However, no serious adverse effects threatening vital signs or permanently decreasing vision were reported.

### 3.4. Heterogeneity and Publication Bias

Regarding the IOP reduction achieved with LBN, the analyses showed high heterogeneity (all *I*^2^ > 93%). When comparing the IOP-reduction effects between LBN and timolol or latanoprost, we still found high heterogeneity among the included studies (all *I*^2^ > 80%).

Figure 5 shows the funnel plots for calculating publication bias in studies analyzing the IOP reduction obtained with LBN, comparing IOP reduction between LBN and timolol, and comparing IOP reduction between LBN and latanoprost. The graphics were generally symmetric and were found to be nonsignificant based on Egger’s test (*p* values were 0.13, 0.44 and 0.09, respectively), revealing no significant publication bias in any of the included studies.

## 4. Discussion

This meta-analysis included nine studies (2389 patients with open-angle glaucoma and ocular hypertension) focused on the IOP-reducing efficacy of LBN 0.024%. LBN 0.024% significantly decreased IOP at all time points from 2 weeks to 12 months. Moreover, compared with timolol maleate 0.5% and latanoprost 0.005%, LBN 0.024% was significantly superior in reducing IOP (LBN vs. timolol: SMD −0.61 at 2 weeks, −0.66 at 6 weeks, and −0.98 at 3 months; LBN vs. latanoprost: SMD −0.599 pooled over each time point).

LBN breaks down into two active components, latanoprost and a NO-donating agent, on the ocular surface which can increase aqueous humor outflow through the uveoscleral and trabecular routes, respectively [18,19]. LBN has an additional NO-donating property, thus explaining why LBN reduces IOP more than latanoprost and timolol.

NO and its second messenger, cyclic guanosine monophosphate (cGMP), mediate smooth muscle relaxation and vasodilation [20]. Similarly, NO exerts action on trabecular meshwork cells, which are highly contractile, and reduces their cell volume [21]. NO may also relax the inner wall of Schlemm’s canal and disrupt intercellular adherens junctions, thus enhancing the outflow from the trabecular route [22,23]. In in vitro studies and animal studies, LBN demonstrated a superior hypotensive effect compared to latanoprost [6,24]. Our meta-analysis, which was comprehensive, enrolling clinical trials and real-world studies in humans regarding LBN to date, found that LBN has a better IOP-lowering effect than latanoprost 0.005% and timolol maleate 0.5%. Our findings were compatible with those in a previous meta-analysis conducted by Harasymowycz et al. [25] evaluating the short-term treatment efficacy at 3 months. The strength of our study is the longer study period (up to 12 months). Therefore, we summarize the mid-term efficacy of LBN 0.024%. Another strength of our study is that we analyzed the IOP-lowering effect at various time points. We noticed that after 2 weeks of treatment, the level of the IOP reduction was stable to the end of follow-up (12 months) without prominent fluctuation. Further long-term studies are needed to understand the long-term benefit of LBN 0.024%.

Another strength of our study is that the patients we enrolled were naive to glaucoma eye drops or had experienced a washout period; thus, we could derive the pure effect of LBN 0.024%. However, in most real-world practice, LBN 0.024% is added to the patient’s regimen or the patient is switched to LBN. Therefore, further real-world studies are warranted to assess the IOP-lowering effect in patients with add-on or switching to LBN 0.024%.

One limitation of our analyses is the high heterogeneity among the included studies. The heterogeneity might originate from diversity in the study population, such as demographics, the severity of glaucoma, baseline IOP, and compliance. Fortunately, despite the heterogeneity, all of the studies demonstrated a tendency towards a significant IOP reduction, thereby providing strong evidence of the effectiveness of LBN 0.024%. Another limitation is that only nine studies were included in our meta-analysis. Treatment effects of LBN 0.024% and the other two drugs were compared within a short-term, three-month duration. With the approval and commercialization of LBN 0.024%, more studies will be conducted to evaluate the long-term effectiveness.

We have found that LBN 0.024% was generally safe and had only minor side effects. Given that LBN 0.024% is among the first NO-donating compounds for topical ophthalmic use, we should continue monitoring the possibility of long-term side effects. Furthermore, previous studies have suggested that NO enhances ocular perfusion and thus possesses a neuroprotective function [26,27]. Due to limited data at present, we cannot conduct a meta-analysis regarding the neuroprotective effects. Further research will be warranted to evaluate the impact of LBN 0.024% on the preservation of the nerve fiber layer or visual field, thus providing profound knowledge of LBN 0.024%.

## 5. Conclusions

Our study used a meta-analysis to investigate the IOP-lowering performance of LBN 0.024%. According to a pooled analysis of 2389 patients with OAG or OHT from nine trials, we found that LBN 0.024% could significantly reduce IOP and was more effective than timolol maleate 0.5% and latanoprost 0.005% during the follow-up period of three months. There were no serious adverse effects related to LBN.

## Figures and Tables

**Figure 1 jcm-11-04325-f001:**
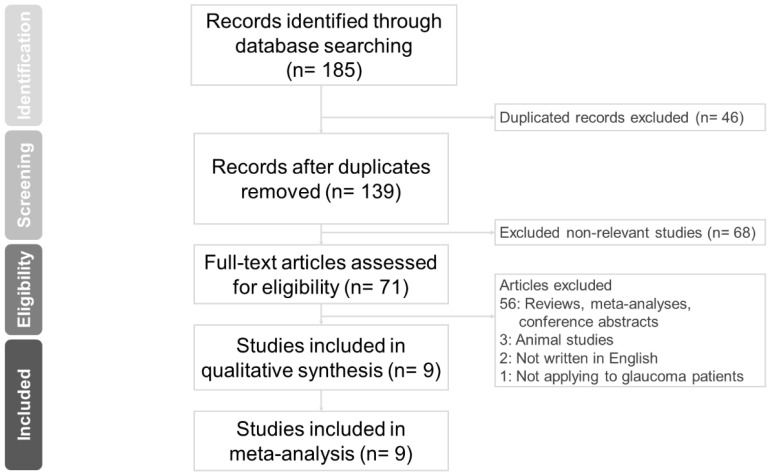
Study retrieval process according to the Preferred Reporting Items for Systematic Reviews and Meta-Analysis (PRISMA) statement.

**Figure 2 jcm-11-04325-f002:**
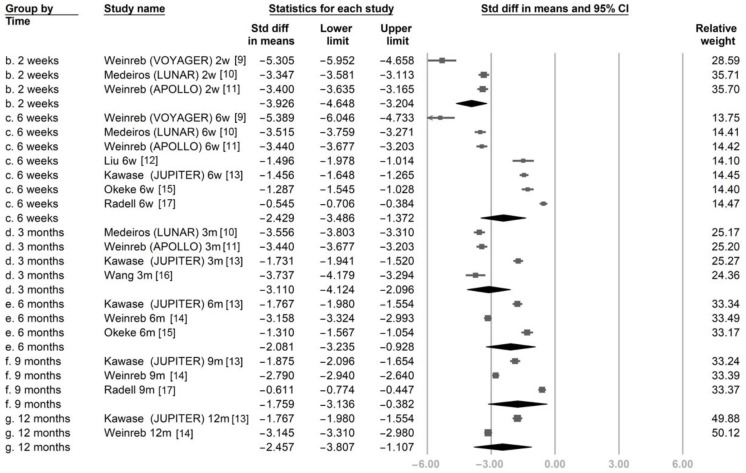
The overall effect of latanoprostene bunod on intraocular pressure.

**Figure 3 jcm-11-04325-f003:**
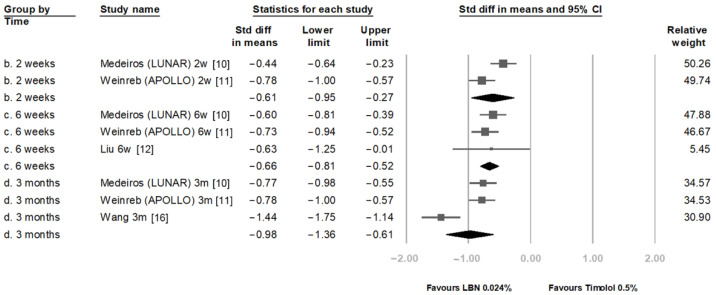
Comparison of IOP-lowering effect between latanoprostene bunod 0.024% and timolol maleate 0.5%.

**Figure 4 jcm-11-04325-f004:**
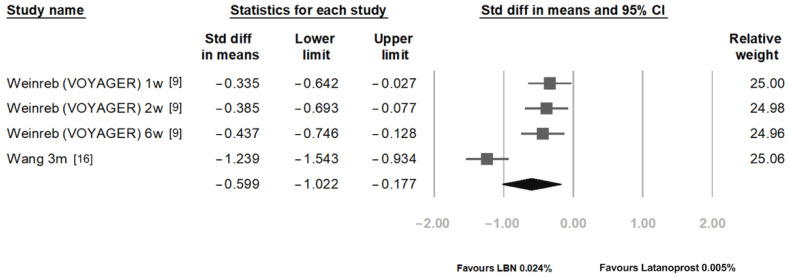
Comparison of IOP-lowering effects of latanoprostene bunod 0.024% and latanoprost 0.005%.

**Figure 5 jcm-11-04325-f005:**
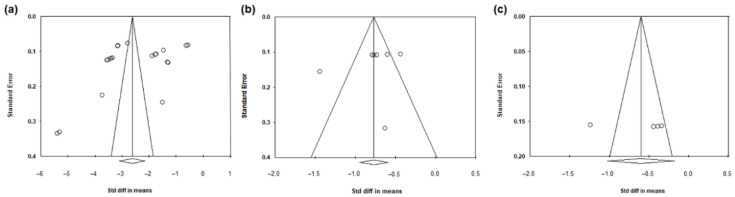
Funnel plots evaluating the publication biases regarding the IOP-lowering impacts of (**a**) LBN 0.024%; (**b**) the comparison between LBN 0.024% and timolol maleate 0.5%; and (**c**) the comparison between LBN 0.024% and latanoprost 0.005%. Circles represented studies.

**Table 1 jcm-11-04325-t001:** Demographic characteritics of patients in studies included in meta-analysis.

First Author	Year	Country	Study Design	Diagnosis	Groups	Num of pts	Num of Eyes	Age (Mean ± SD)	Male, n (%)
Weinreb (VOYAGER) [9]	2015	USA, Europe	Multicenter RCT	POAG, OHT	LBN	83	83	60.8 ± 11.47	26 (31.3)
					Latanoprost	82	82	61.2 ± 11.92	29 (35.4)
Medeiros (LUNAR) [10]	2016	USA, Europe	Multicenter RCT	POAG, OHT	LBN	259	278	65.0 ± 9.77	116 (41.7)
					Timolol	128	136	64.1 ± 9.71	57 (41.9)
Weinreb (APOLLO) [11]	2016	USA, Europe	Multicenter RCT	POAG, OHT	LBN	284	284	64.7 ± 10.3	118 (41.5)
					Timolol	133	133	63.1 ± 11.2	56 (42.1)
Liu (CONSTELLATION) [12]	2016	USA	Prospective, open-label, RCT	POAG, OHT	LBN	25	25	60.3 ± 10.6	7(28)
					Timolol	25	25	60.3 ± 10.6	7(28)
Kawase (JUPITER) [13]	2016	Japan	single-arm, multicenter, open-label, clinical study	OAG, OHT	LBN	130	130	62.5 ± 18.9	56(43.1)
Weinreb (Pooled) [14]	2018	USA, Europe	Pooled analysis of APOLLO and LUNAR	POAG, OHT	LBN	562	562	64.9 ± 10.04	234 (41.6)
					Timolol *	269	269	63.7 ± 10.47	113 (42.0)
Okeke [15]	2020	USA	Multi-center retrospective	POAG, LTG, others	LBN	65	65	59.3 ± 14.4	30 (46.2)
Wang [16]	2020	China	Retrospective	POAG, OHT	Latanoprost	104	NR	58.42 ± 6.12	55 (53)
					LBN	94	NR	57.65 ± 6.01	49 (52)
					Timolol	115	NR	57.99 ± 6.44	53 (46)
Radell [17]	2021	USA	Single center retrospective	POAG, LTG, others	LBN	56	102	68.8 ± 12.4	28 (50)

Num number, pts patients, SD Standard deviation, NR Not-reported, RCT Randomized controlled trial, POAG Primary open-angle glaucoma, OHT Ocular hypertension, LTG low tension glaucoma, LBN Latanoprostene bunod. * crossover to LBN after 12 weeks.

**Table 2 jcm-11-04325-t002:** IOP at baseline and post-treatment visits.

			Post-Treatment IOP						
Study	Groups	Baseline IOP (Mean ± SD)	1 Week	2 Weeks	6 Weeks	12 Weeks	6 Months	9 Months	12 Months
Weinreb (VOYAGER), 2015 [9]	LBN	26.01 ± 1.67	17.74	17.15	17.01	NR	NR	NR	NR
	Latanoprost	26.15 ± 1.79	18.86	18.43	18.38	NR	NR	NR	NR
Medeiros (LUNAR), 2016 [10]	LBN	26.6 ± 2.39	NR	18.6	18.2	18.1	NR	NR	NR
	Timolol	26.4 ± 2.30	NR	19.2	19.1	19.3	NR	NR	NR
Weinreb (APOLLO), 2016 [11]	LBN	26.7 ± 2.5	NR	18.2	18.1	18.1	NR	NR	NR
	Timolol	26.5 ± 2.4	NR	19.5	19.3	19.4	NR	NR	NR
Liu (CONSTELLATION), 2016 [12]	LBN	21.6 ± 2.8	NR	NR	17.6 ± 2.5	NR	NR	NR	NR
	Timolol	21.6 ± 2.8	NR	NR	18.9 ± 2.4	NR	NR	NR	NR
Kawase (JUPITER), 2016 [13]	LBN	19.6 ± 2.9	NR	NR	15.3 ± 3.0	14.8	14.7	14.4	14.7
Weinreb (Pooled), 2018 [14]	LBN *	26.7 ± 2.43	NA	NA	NA	NA	18.1 ± 2.9	18.2 ± 3.3	17.9 ± 3.0
Okeke, 2020 [15]	LBN	21.7 ± 5.9	NR	NR	14.7 ± 4.1	NR	14.4 ± 3.2	NR	NR
Wang, 2020 [16]	Latanoprost	24.13 ± 1.12	NR	NR	NR	19.45 ± 1.01	NR	NR	NR
	LBN	23.98 ± 1.22	NR	NR	NR	17.45 ± 1.89	NR	NR	NR
	Timolol	24.39 ± 1.65	NR	NR	NR	19.68 ± 1.08	NR	NR	NR
Radell, 2021 [17]	LBN	16.2 ± 4.3	NR	NR	14.0 ± 3.6	NR	NR	13.7 ± 3.8	NR

IOP Intraocular pressure, SD Standard deviation, w week, m month, LBN Latanoprostene bunod, NR Not-reported, NA Not-applicable. * Extension phases (start from month 3).

**Table 3 jcm-11-04325-t003:** Treatment-related ocular adverse effects in studies included in meta-analysis.

Study	Groups	Num of Eyes	≥1 AE, n (%)	Types of Complication (%)
Weinreb(VOYAGER), 2015 [9]	LBN	83	20 (24.1)	Ocular hyperaemia (2.4), Conjunctival hyperaemia (4.8), Eye irritation (3.6), Punctate keratitis (2.4), Dry eye (2.4), Photophobia (2.4), Instillation site pain(12)
	Latanoprost	82	10 (12.2)	Ocular hyperaemia (8.5), Punctate keratitis (1.2), Instillation site pain(6.1)
Medeiros(LUNAR), 2016 [10]	LBN	277	66 (23.8)	Conjunctival hyperemia(9), Eye irritation (7.2), Eye pain (5.8), Ocular hyperemia (2.5), Vision blurred (1.8), Eye pruritis (1.4), Dry eye (1.1), Punctate keratitis (1.1), Foreign body sensation (1.1), Instillation site pain(1.4)
	Timolol	135	18 (13.3)	Conjunctival hyperemia(0.7), Eye irritation (4.4), Eye pain (3.7), Ocular hyperemia (0.7), Vision blurred (2.2), Eye pruritis (0.7), Dry eye (0.7)
Weinreb(APOLLO), 2016 [11]	LBN	283	38 (13.4)	Eye irritation (3.9), Conjunctival hyperemia (2.8), Eye pain (1.4), Dry eye (1.1), Foreign body sensation (1.1), Instillation site pain (1.1)
	Timolol	135	16 (11.9)	Eye irritation (2.2), Conjunctival hyperemia (1.5), Eye pain (2.2), Dry eye (0.7), Instillation site pain (1.5)
Liu, 2016 [12]	LBN	23	2(8.7)	Punctate keratitis (4.3) Instillation site erythema (4.3)
	Timolol	23	4(17.4)	Punctate keratitis (13), Instillation site irritation (4.3)
Kawase (JUPITER), 2016 [13]	LBN	130	76 (58.5)	conjunctival hyperemia (17.7), growth of eyelashes (16.2 ), eye irritation (11.5), eye pain (10 ), Iris hyperpigmentation(3.8), Blepharal pigmentation(3.1), Blepharitis (2.3), Eye pruritus(2.3), Asthenopia(2.3), Conjunctival hemorrhage (1.5), Punctate keratitis(2.3), Trichiasis(2.3), Cataract(0.8), Hordeolum(0.8), Visual impairment (0.8), Vitreous floaters (0.8), Foreign body sensation(1.5)
Weinreb (Pooled), 2018 [14]	LBN(LBN + Timolol cross over to LBN)	811	175 (21.6)	Conjunctival hyperemia(5.9), Eye irritation (4.6), Eye pain (3.6), Ocular hyperemia (2), Instillation site pain (2)
	Timolol	271	34 (12.5)	Conjunctival hyperemia(1.1), Eye irritation (2.6), Eye pain (2.2), Ocular hyperemia (0.7), Instillation site pain (1.8)
Okeke, 2020 [15]	LBN	65	33 (50.8)	Blurred vision (15.4), Dryness (12.3), Irritation(7.7), Itching (7.7), Light sensitivity (7.7), Burning (6.2), Eye pain (4.6), Tearing (4.6), Change in vision (3.1), Keratitis (3.1), Macular degeneration (3.1)
Wang, 2020 [16]	Latanoprost	104	NR	Eye irritation (4), Dry eye(2), Eye pain(3), Conjunctival hyperemia(4), Foreign body sensation(3)
	LBN	94	NR	Eye irritation (4), Dry eye(2), Eye pain(3), Conjunctival hyperemia(3), Foreign body sensation(2)
	Timolol	115	NR	Eye irritation (2), Dry eye(1), Eye pain(1), Conjunctival hyperemia(1), Foreign body sensation(1)
Radell, 2021 [17]	LBN	56	8(14)	pain, itching (14)

Num number, AE treatment-related ocular adverse effects, LBN Latanoprostene Bunod.

## Data Availability

Data analyzed in this study were a re-analysis of existing data which are openly available at locations cited in the reference section.

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
