# Peer review of "Latanoprostene Bunod 0.024% in the Treatment of Open-Angle Glaucoma and Ocular Hypertension: A Meta-Analysis"

_jcm, 2022, doi:10.3390/jcm11154325_

Round 1

Reviewer 1 Report

This article is mainly about a meta-analysis of Latanoprostene bunod 0.024% in the treatment of open-angle glaucoma and ocular hypertension. It contributed to reveal that LBN 0.024% could significantly reduce IOP and was more effective than timolol maleate 0.5% and latanoprost 0.005% according to a pooled analysis of 2389 patients with OAG or OHT from 9 trials. However, there are several mistakes that need to be corrected. Overall, this is an interesting article which is worthy of consideration after major revise.

Major concerns related to this paper:

1. The line of 109-111 in Page 3 is: We further excluded six papers not written in English, two animal studies, and one study in which LBN was not used in glaucoma patients.” The sentence is inconsistent with Figure1.

2. The line of 154-155 in Page 6 is: Comparison of IOP-lowering effects of latanoprostene bunod 0.024% and latanoprost 0.005%.” But it was shown as timolol 0.5% rather than latanoprost 0.005% in Figure 4.

3. The picture (c) in Figure 5 showed a P-value of 0.09, but it showed a P-value of 0.08 in the line of 164 in Page 6.

Reviewer 2 Report

This is a meta-analysis on Latanoprostene bunod 0.024% in the treatment of open-angle glaucoma and ocular hypertension. The search strategy is to include studies on the effect of IOP lowering in LBN with or without comparison to maleate 0.5% or latanoprost 0.005%. Overall, the logic of this article is clear. However, a similar meta-analysis has recently been published (Br J Ophthalmol. 2022;106(5):640-647.). This article does not provide much valuable new information.

My suggestion is to do an in-depth analysis of the side effects of the drug and the potential protective effects of the optic nerve independent of IOP, rather than just mention them in the discussion.

In addition, according to Table 2 of the manuscript, all studies compared with timolol or latanoprost were only observed for 3 months, so in the conclusion, LBN 0.024% was more effective than timolol maleate 0.5% and latanoprost 0.005%,  It is not accurate and need an attribute.

Reviewer 3 Report

The authors review includes the results of the efficacy of Latanoprost and Timolol with Latanoprostene. Undoubtedly, the research is relevant.  Any development in the line of ocular hypotensive medication with prostaglandins and prostanoids is relevant. The review shows data from different series of ocular antiglaucoma medications. The topic is original. The review is well-structured and easy to read. The article is well written and the conclusions are consistent and answer the main question.

Round 2

Reviewer 2 Report

The revisions improved the manuscript.